# Rare Earth Elements Enhanced the Oxidation Resistance of Mo-Si-Based Alloys for High Temperature Application: A Review

**Laihao Yu, Yingyi Zhang \*, Tao Fu, Jie Wang, Kunkun Cui and Fuqiang Shen**

School of Metallurgical Engineering, Anhui University of Technology, Maanshan 243002, China; aa1120407@126.com (L.Y.); ahgydxtaofu@163.com (T.F.); wangjiemaster0101@outlook.com (J.W.); 15613581810@163.com (K.C.); sfq19556630201@126.com (F.S.)
\* Correspondence: zhangyingyi@cqu.edu.cn; Tel.: +86-17375076451

**Abstract:** Traditional refractory materials such as nickel-based superalloys have been gradually unable to meet the performance requirements of advanced materials. The Mo-Si-based alloy, as a new type of high temperature structural material, has entered the vision of researchers due to its charming high temperature performance characteristics. However, its easy oxidation and even "pesting oxidation" at medium temperatures limit its further applications. In order to solve this problem, researchers have conducted large numbers of experiments and made breakthrough achievements. Based on these research results, the effects of rare earth elements like La, Hf, Ce and Y on the microstructure and oxidation behavior of Mo-Si-based alloys were systematically reviewed in the current work. Meanwhile, this paper also provided an analysis about the strengthening mechanism of rare earth elements on the oxidation behavior for Mo-Si-based alloys after discussing the oxidation process. It is shown that adding rare earth elements, on the one hand, can optimize the microstructure of the alloy, thus promoting the rapid formation of protective $SiO_2$ scale. On the other hand, it can act as a diffusion barrier by producing stable rare earth oxides or additional protective films, which significantly enhances the oxidation resistance of the alloy. Furthermore, the research focus about the oxidation protection of Mo-Si-based alloys in the future was prospected to expand the application field.

**Keywords:** Mo-Si-based alloys; alloying; rare earth elements; oxidation behavior; mechanism





## 1. Introduction

As the world population increases, the problem of global resource shortage has become increasingly prominent. It is well known that in addition to waste recycling, improving energy utilization and exploring new energy are effective methods to solve resource problems, and are also the main trend of future scientific and technological development [1,2]. Nowadays, the development of new energy is overwhelming, which urges scientists to study the properties of materials on a smaller and smaller scale, so that it is easier to develop new materials [3,4]. For example, Ghidelli and Ast et al. [5–7] systematically studied the mechanical properties of ceramics at the submicrometre scale such as fracture toughness, and improved the recently developed pillar splitting method that can provide reliable and simple ways to measure fracture toughness over a broad range of material properties. As a new high temperature structural material, the Mo-Si-based alloy is expected to replace the nickel-based alloy and play an important role in turbine engine and industrial furnace components [8,9].

A large number of studies have pointed out that Mo-Si-based alloys have outstanding high temperature performance characteristics, such as moderate density, strong electrical and thermal conductivity, ultra-high melting point, high thermal impact resistance, etc., which have been widely used in various industries [10,11]. However, these alloys also

have some inherent defects that limit their generalization as structural materials [12–14]. For example, $MoSi_2$-based alloys exhibit low room temperature fracture toughness, poor high temperature creep resistance, and catastrophic oxidation at 400 to 800 °C. Although $Mo_5Si_3$-based alloys have relatively strong creep resistance, they generally present an accelerated oxidation or "pesting oxidation" phenomenon below 1000 °C, which is a thorny problem [15–19]. To overcome these shortcomings, relevant research work has not stopped since the end of the 20th century [20–25]. Fortunately, people finally succeeded in improving the properties of alloy through material designs [26–29], preparation techniques [30–34] or surface modification methods [35–38]. Among them, doping second phases such as W, Nb, $ZrO_2$, $La_2O_3$, $Al_2O_3$, and $Cr_2O_3$ in material designs to strengthen the performance of substrate is generally regarded as an important measure [39]. In the case of preparation technique, Ghidelli et al. [40] used a pulsed laser deposition technique to prepare a class of new nano-structured $Zr_{50}Cu_{50}$ (at.%) metallic glass films with excellent and tunable mechanical properties. In the case of surface modification methods, Besozzi et al. [41] studied the thermomechanical properties of different systems of amorphous tungsten–oxygen and tungsten–oxide coatings, which was of great significance to the design and construction of devices.

There is no doubt that in some refractory metal materials like niobium-based, molybdenum-based, and tungsten-based materials, adding active elements may significantly improve the material properties [42–44]. For example, the addition of Si and B to molybdenum-based materials can produce protective borosilicate scales on the substrate surface, which helps to reduce the oxidation rate. Moreover, the presence of B can also facilitate the $SiO_2$ scale flow by reducing the scale viscosity, thereby filling the pores and cracks that may occur on the alloy surface. Rare earth or its oxides, as a kind of active element, have received particular attentions and widespread applications [45–48]. In recent years, researchers have made great breakthroughs in studying rare earth elements to enhance the mechanical properties of metal materials. However, so far, there are few reports on the oxidation behavior of Mo-Si-based alloy doped with rare earth elements. Therefore, this paper comprehensively and systematically reviewed the actions of rare earth elements such as La, Hf, Ce and Y on the antioxidant properties of Mo-Si-based alloys, especially the Mo-Si-B system, and summarized the relevant strengthening mechanisms.

## 2. Effects of Rare Earth Elements on Oxidation Behavior

It's well known that the increase of oxide layer thickness is primarily caused by the internal diffusion of $O_2$ [49–52], and studies have shown that alloying with active elements can effectively reduce the internal diffusion rate of $O_2$ [53–55]. For example, appropriate adding rare earth elements, on the one hand, can separate oxygen atoms at the metal-oxide interfaces and oxide scale grain boundaries and react with $O_2$, thus hindering the diffusion of $O_2$ [56–59]. On the other hand, it can optimize the oxide scale microstructure and improve the scale adhesion [60–63].

### 2.1. Effects of Rare Earth Elements on Mo-Si-B Alloys

Among various Mo-Si-based alloys, Mo-Si-B alloys are the most widely studied [64–70]. Therefore, we first discuss the effect of rare earth elements on the oxidation properties of Mo-Si-B alloys.

### 2.1.1. Effects of La Element Addition

Compared with pure Mo-Si alloys, adding La element can significantly optimize the microstructure of these alloys by the means of reducing the grain size and making the intermetallic particles disperse more evenly, thus improving the fracture toughness, bending and compressive strength significantly [71–74]. Based on the existing studies of Mo-12Si-8.5B alloy (at.%) [75–79], we further analyzed the actions of doping La or $La_2O_3$ s phase on the oxidation behavior.

Zhang et al. [80] prepared Mo-12Si-8.5B (at.%, abbreviated as MSB) samples added with different contents of La$_2$O$_3$ through arc-melted and spark plasma sintered methods, and the specific contents were presented in Table 1. Figure 1a gives the XRD patterns of MSB + xLa$_2$O$_3$ samples (x = 0, 0.3, 0.6, 1.2 wt.%). It can be seen that all the samples consist of Mo$_5$SiB$_2$, Mo$_3$Si and α-Mo three phases, which is consistent with the phase diagram of isothermal Mo-Si-B composites (Figure 1b) [76]. At the same time, it also reveals that even if La$_2$O$_3$ is added, it will not affect the phase composition of samples. Figure 1c–f are micrographs of the four samples prepared, where the white regions are α-Mo phase, and the black regions are Mo$_5$SiB$_2$/Mo$_3$Si phases dispersed in the α-Mo matrix. It can also be found from micrographs that the grain size of α-Mo and Mo$_5$SiB$_2$/Mo$_3$Si will be reduced after adding La$_2$O$_3$, in which the α-Mo size change is more pronounced, whereas the decrease of each phase size is not sensitive to the La$_2$O$_3$ mass fraction. Moreover, the distribution of Mo$_5$SiB$_2$/Mo$_3$Si phases is more uniform after doping La$_2$O$_3$. This is because parts of La$_2$O$_3$ can be used as nucleation sites, which leads to the increase of nucleation density. On the other hand, La$_2$O$_3$ particles play a "pinning" role on the α-Mo boundary to inhibit its grains growth. The results in Table 1 further reveal the effect of La$_2$O$_3$ on grain size.

**Table 1.** The La$_2$O$_3$ mass fractions and grain sizes of various samples studied. Reproduced with permission [80]. Copyright 2011 Elsevier.

| Materials | MSB | MSB + 0.3 | MSB + 0.6 | MSB + 1.2 |
|---|---|---|---|---|
| Mass fraction of La$_2$O$_3$ (wt.%) | 0 | 0.3 | 0.6 | 1.2 |
| Grain sizes of α-Mo (μm) | 19.78 | 10.88 | 9.56 | 9.46 |
| Grain sizes of Mo$_3$Si/Mo$_5$SiB$_2$ (μm) | 3.04 | 2.46 | 2.55 | 2.17 |

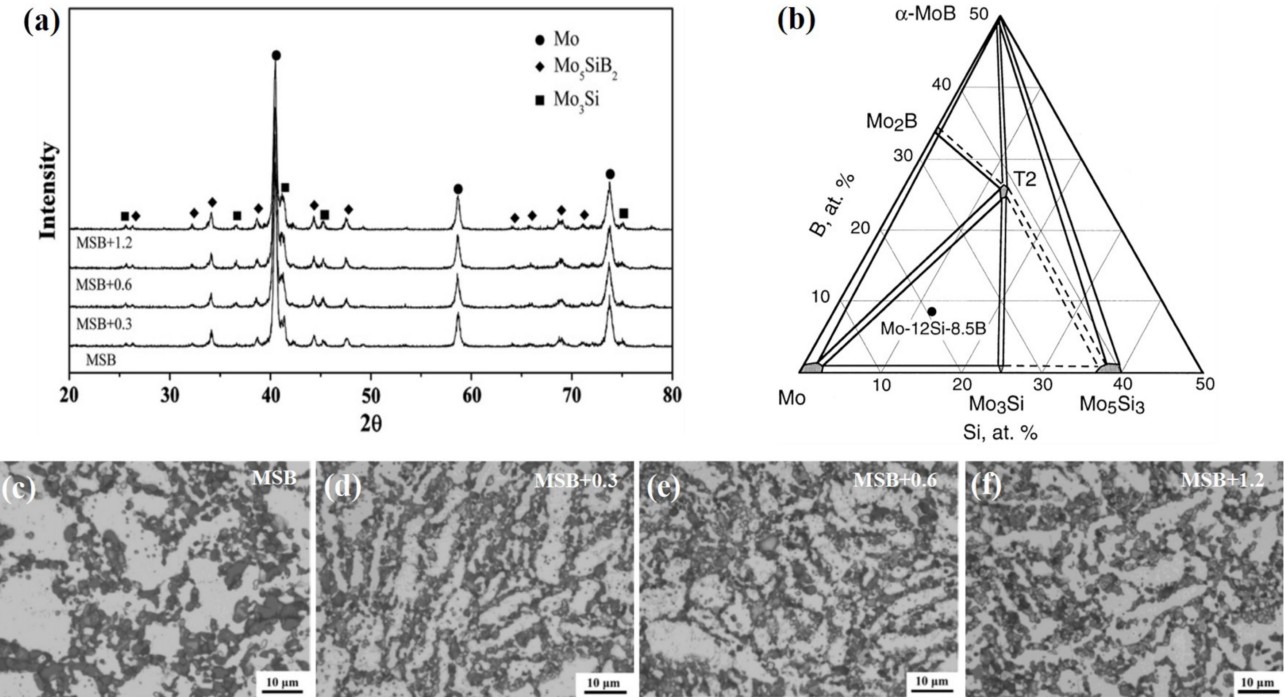

**Figure 1.** XRD patterns (**a**) and microstructure images (**c–f**) of MSB + xLa$_2$O$_3$ samples (x = 0, 0.3, 0.6, 1.2 wt.%) before oxidation; schematic section of the 1600 °C isothermal ternary Mo-Si-B phase diagram (**b**). (**a**,**c–f**) and (**b**) reproduced with permission from [80] and [76], respectively. Copyright 2011 Elsevier and 2001 Elsevier.

Weight variation curves for different mass fraction La$_2$O$_3$-doped MSB samples after oxidation at 800 °C are shown in Figure 2a. The results point out that the mass loss of alloy can be significantly reduced after adding La$_2$O$_3$, where MSB + 0.6 sample exhibits the least

mass loss during transient oxidation stage. It is due to the fact that $La_2O_3$-doped MSB samples present a finer grain size, which makes it faster to form a protective borosilicate scale to prevent further volatilization of $MoO_3$. The role of grain refinement has also been reported elsewhere [81–87]. Again, $La_2O_3$ in the alloy can decrease the grain boundary transport rate, leading to the reduction of weight loss rate. To determine the impact of $La_2O_3$ on the oxidation behavior, Zhang et al. [80] further studied the cross-sectional structure of MSB + 0.6 sample oxidized at 800 °C. It has been noticed from Figure 2d that the cross section of this sample is composed of oxidation scale, interlayer and substrate. Among them, the top layer is the dense $B_2O_3$-$SiO_2$ scale and the interlayer comprised with $MoO_2$, which is confirmed by XRD analysis and the content of each element (i.e., Mo, Si, B, O). Furthermore, compared with the MSB sample, the intensity and peaks of $B_2O_3$ and $SiO_2$ of MSB + 0.6 sample are raised visibly (Figure 2b), indicating that the antioxidant capacity of the MSB + 0.6 sample is enhanced. Burk [88,89] and Jéhanno [90] et al. also reported similar results.

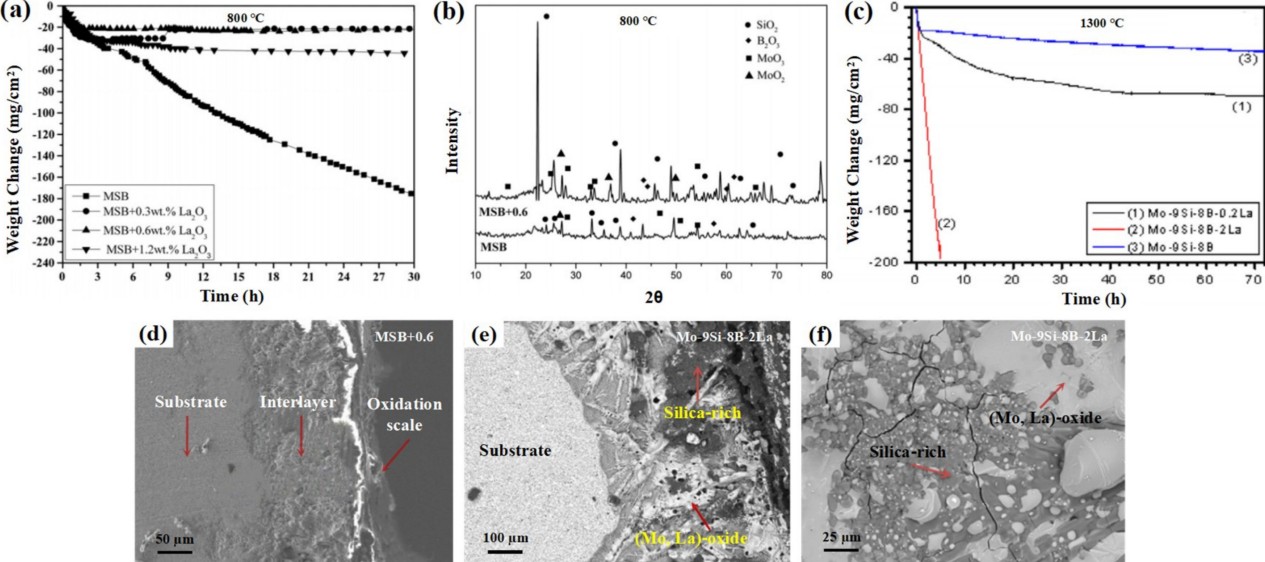

**Figure 2.** Weight change curves of various samples oxidized at different temperatures (**a**,**c**); surface and cross-sectional SEM/BSE images of various samples oxidized at different conditions: 800 °C (**d**) with corresponding surface XRD analysis results (**b**), 1200 °C for 2 h (**e**), and 1300 °C for 23 h (**f**). (**a**,**b**,**d**) and (**c**,**e**,**f**) reproduced with permissions from [80] and [60], respectively. Copyright 2011 Elsevier and 2014 Elsevier.

In addition, Majumdar et al. [59,60] also investigated the oxidizability of the Mo-9Si-8B (at.%) sample doped with 2 at.% La at 750–1400 °C. It is established that the La-doped sample exhibits relatively good antioxidant capability below 1000 °C, which is the result of stable lanthanum oxides like $3La_2O_3 \cdot MoO_3$, $La_2O_3$ and $La_2O_3 \cdot 3MoO_3$ produced at the oxidation scale to inhibit the formation and volatilization of $MoO_3$. However, when the temperature exceeded 1000 °C, the addition of La might adversely affect the sample oxidation properties. Figure 2c displays the weight change curves of La-doped and undoped samples at 1300 °C, and it can be seen that the weight loss of La-doped sample is significantly higher than that of undoped sample. This is because adding La makes the sample cross section present a loose and porous oxide layer structure when oxidized at 1200 °C (Figure 2e). Meanwhile, as oxidation temperature rises to 1300 °C, a large number of cracks and holes are observed on the sample surface (Figure 2f), which provides a pathway for $O_2$ internal diffusion and $MoO_3$ volatilization. In contrast, even if undoped sample is oxidized at 1300 °C for 72 h, a continuous and compact oxidation scale can be still observed in its cross section [88]. Thus, undoped sample has better antioxidant properties in high-temperature environments.

### 2.1.2. Effects of Hf Element Addition

Extensive experiments have reported that adding $Hf/HfB_2$ to Mo-Si-B composites can clearly improve their performance features, such as high temperature strength, high temperature stability, creep resistance, fracture toughness, etc. [91–95]. Potanin et al. [96] discussed the oxidation behavior of $MoB-HfB_2-MoSi_2$ composites at 1200 °C in detail. The composition of each alloy is illustrated in Table 2, where the difference between $X34_2$ and $X34_1$ samples is that the former presents a two-level structure (TLS), while the latter presents a single-level structure (SLS). The microstructures of the studied samples are depicted in Figure 3a–c, and on the whole, the three samples all contain MoB and $MoSi_2$ phases. The difference is that $X34_1$ and $X34_2$ samples also have additional $HfSiO_4$ and $HfB_2$ phases and their grain sizes are finer than that of the X0 sample. Figure 4 gives the oxidation kinetics curves of the three samples, and it can been observed that the weight increases of $X34_1$ and $X34_2$ samples are more obvious because adding $HfB_2$ can make the samples generate $HfSiO_4$ (6.97 $g/cm^3$) and $HfO_2$ (9.68 $g/cm^3$), whose specific weights are greater than $SiO_2$ (2.36 $g/cm^3$) [97]. By the way, the weight gain of the $X34_1$ sample is smaller than that of the $X34_2$ sample. This is because $HfB_2$ and MoB grains in the $X34_2$ sample present finer size and more uniform distribution than the $X34_1$ sample (Figure 3b,c), which causes the more intense oxidation of $X34_2$ sample.

**Table 2.** The elemental composition of various samples. Reproduced with permission [96]. Copyright 2019 Elsevier.

| Samples | Composition (at.%) | | | |
| --- | --- | --- | --- | --- |
| | **Mo** | **Hf** | **Si** | **B** |
| X0 | 35.0 | – | 60.0 | 5.0 |
| $X34_1$ $X34_2$ | 23.2 | 11.3 | 39.7 | 25.8 |

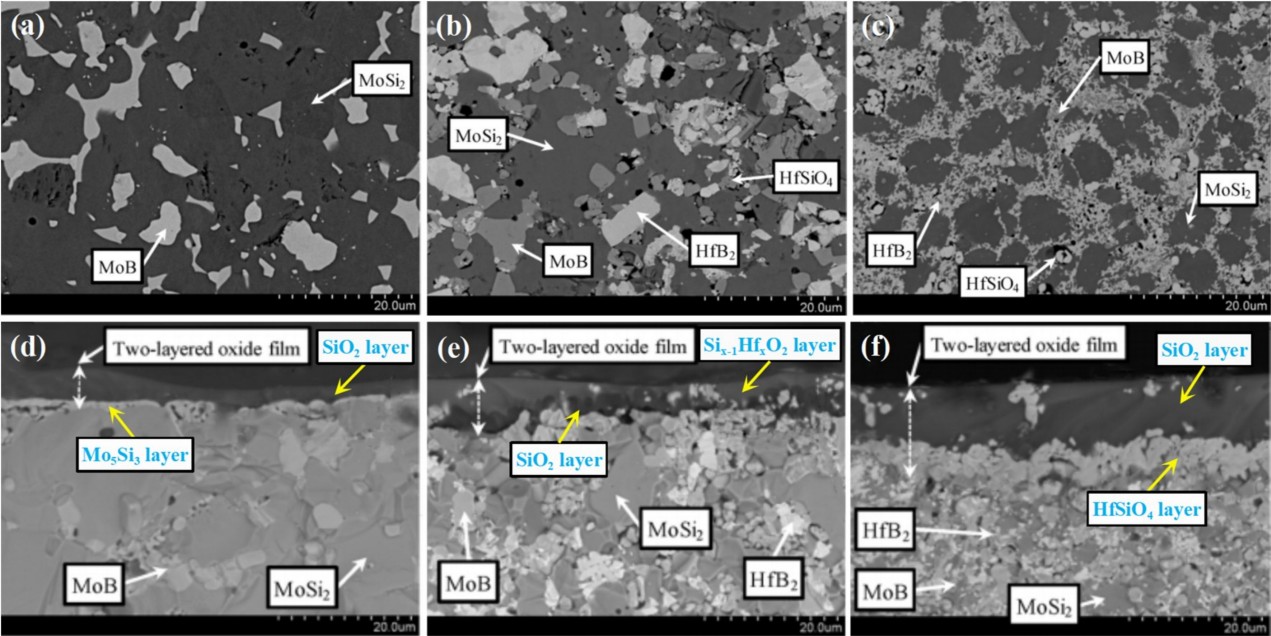

**Figure 3.** Microstructure of various samples: (**a**) X0, (**b**) $X34_1$ and (**c**) $X34_2$; cross-sectional SEM images of samples after oxidation at 1200 °C for 30 h: (**d**) X0, (**e**) $X34_1$ and (**f**) $X34_2$. Reproduced with permission [96]. Copyright 2019 Elsevier.

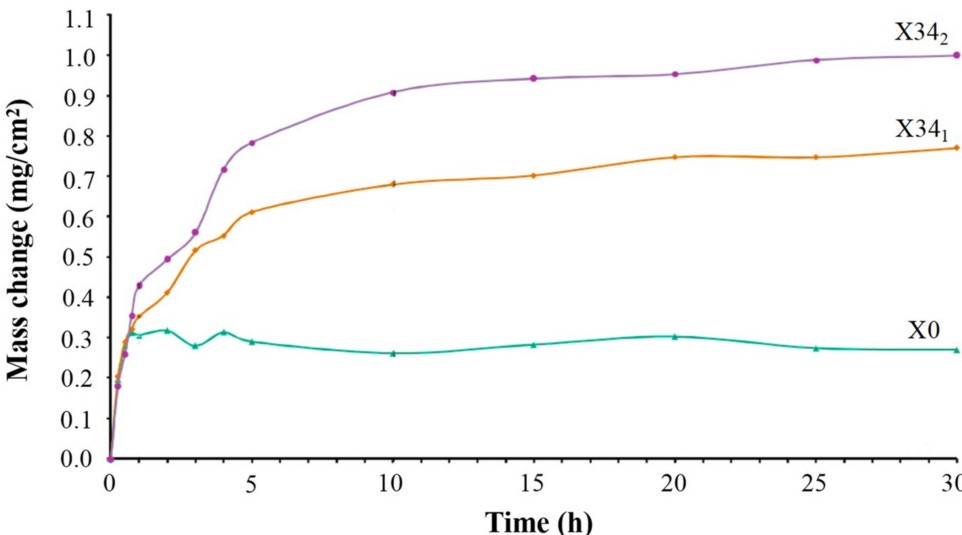

**Figure 4.** Kinetic curves of the samples oxidized at 1200 °C. Reproduced with permission [96]. Copyright 2019 Elsevier.

Figure 3d–f shows the cross-sectional structure of X0, $X34_1$ and $X34_2$ samples after oxidation at 1200 °C for 30 h. It has been noticed that the $X34_1$ and $X34_2$ samples do produce $Si_{x-1}Hf_xO_2$/$HfSiO_4$ phases during the oxidation process, which is consistent with the above analysis results. Meanwhile, two-layered oxide films are formed on the surface of all samples, whereas there are great differences in the oxide-film composition and structure. In other words, the X0 sample oxide scale containing the $SiO_2$ layer (top layer) and $Mo_5Si_3$ layer (bottom layer), the $X34_1$ sample oxide scale consists of a $Si_{x-1}Hf_xO_2$-doped amorphous $SiO_2$ layer (upper layer) and a crystalline $\alpha$-$SiO_2$ layer (lower layer), while the $X34_2$ sample oxide scale is comprised of a crystalline $\alpha$-$SiO_2$ layer (outermost layer) and a $HfSiO_4$ layer (interlayer). Figure 5 gives the TEM images of $X34_1$ sample after oxidation. It can be seen that two $SiO_2$-based layers have been formed on the sample surface: crystalline lower layer and amorphous upper layer, which is close to the unoxidized $MoSi_2$ ceramic. At the same time, the EDS analysis shows that the upper layer contains $2.4 \pm 0.7$ at.% Hf in addition to O and Si. It is speculated that the amorphization of $SiO_2$ may be related to the dissolution of Hf. As Hf and Si are equivalent elements, Hf can be incorporated into $SiO_2$ lattice to form $Si_{x-1}Hf_xO_2$ solid solution and suppress the transition of $SiO_2$ from amorphous to the crystalline state upon heating. Meanwhile, the Si–O–Si bond is stretched owing to the higher atomic radius of Hf (0.167 nm) than Si (0.132 nm). Furthermore, the amorphous phenomenon of $SiO_2$ may also be associated with the oxidation and cooling rate of $X34_1$ sample. It is worth noting that the $X34_2$ sample oxide scale exhibits denser structure is caused by the existence of high wetting angle makes $SiO_2$-$B_2O_3$ melt can shrink $HfSiO_4$ particles together, resulting in the formation of smooth and compact oxide films [98].

To summarize, addition of $HfB_2$ to X0 sample can produce $HfSiO_4$/$Si_{x-1}Hf_xO_2$ particles dispersed in oxide film, and even form an $HfSiO_4$ interlayer. It has been proven that the $HfSiO_4$ and $ZrSiO_4$ particles have similar effects. On the one hand, they can promote the healing of cracks and holes in the borosilicate scale [99,100], on the other hand, they can both act as barriers and $HfSiO_4$ particles can also increase the crystallization temperature of amorphous scale [101]. Therefore, adding $HfB_2$ can enhance the alloy antioxidant effects. The research results of Sciti et al. also confirmed this conclusion [97].

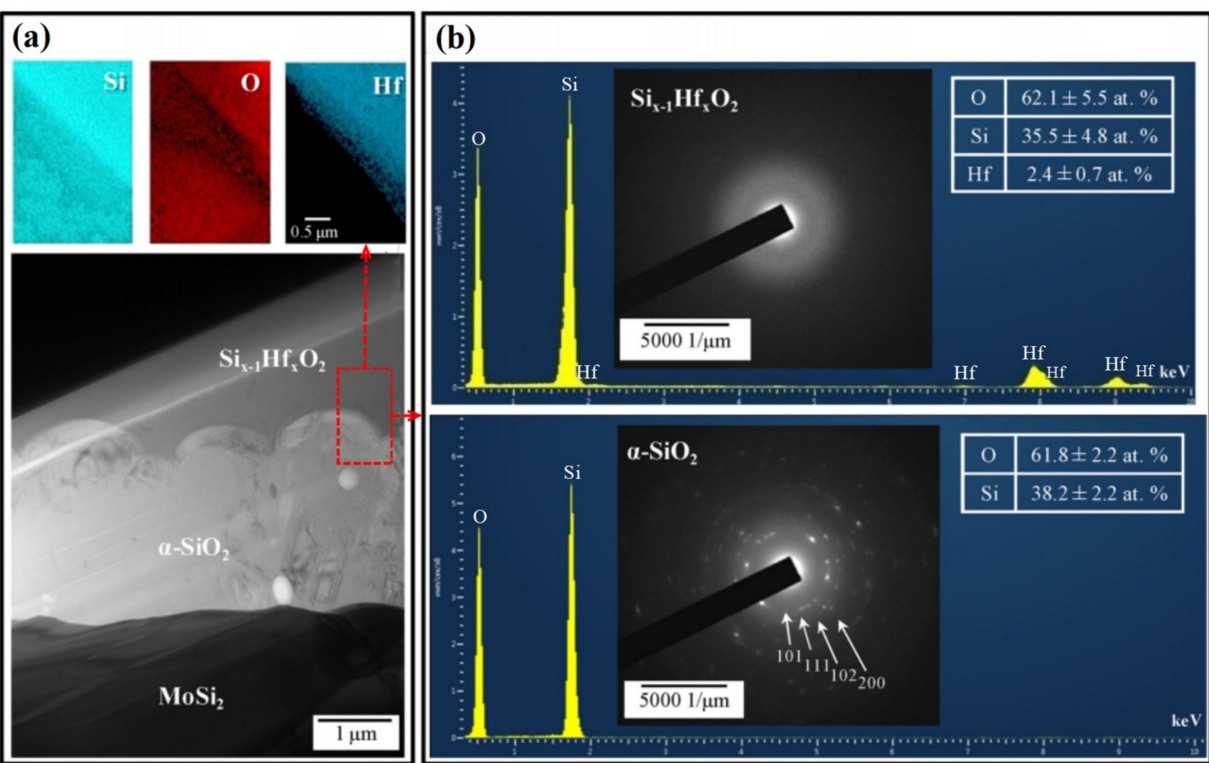

**Figure 5.** (**a**) TEM micrograph of the oxidized X34$_1$ sample; (**b**) EDS and diffraction patterns of the oxide scale. Reproduced with permission [96]. Copyright 2019 Elsevier.

### 2.1.3. Effects of Ce Element Addition

Actions of Ce on oxidation performance of Mo-Si-B alloys have been investigated deeply. Das et al. [102,103] performed isothermal oxidation experiments about Mo-11.18Si-8.09B-7.29Al-0.16Ce, Mo-11.2Si-8.1B-7.3Al, Mo-13.98Si-9.98B-0.16Ce and Mo-14Si-10B alloys (at.%, abbreviated as MSBACe, MSBA, MSBCe and MSB$_1$, respectively) synthesized by arc melting. The experiment results have suggested that doping a small amount of Ce has little effect on oxidation kinetics of the alloy at 500 and 700 °C (Figure 6a), while the presence of Ce presents a palpable effect on the alloy oxidation behavior at 900–1300 °C. Figure 6b–d provide the mass variation curves of MSB$_1$ and MSBCe oxidized at 900–1300 °C, respectively. It can be seen that the mass loss of MSB$_1$ increases after the addition of Ce (Figure 6c,d). Even so, MSBCe exhibits shorter transient oxidation periods as compared to MSB$_1$, and its steady-state stage curve almost presents a horizontal trend, revealing that MSBCe is more effectively protected than MSB$_1$.

At the same time, the microstructural morphology of two oxidized alloys is shown in Figure 7. During oxidation at 900 °C, one-layered oxide film (i.e., Mo-oxide film) and flowing glassy phase are observed on the surface of both alloys (Figure 7a,d). The difference is that the glassy phase in MSBCe flows faster due to the presence of Ce, hence MSBCe presents a smaller mass variation at 900 °C (Figure 6b). However, after oxidation at 1100 °C for 24 h, the oxide-layer structure of both alloys has changed distinctly, namely SiO$_2$ layer is formed on top of the Mo-oxide layer (Figure 7b,e). When the oxidation temperature reaches 1300 °C, B$_2$O$_3$ has begun to evaporate from the oxide scale, resulting in increased viscosity and weak fluidity of the scale, which leads to the deterioration of alloy antioxidation ability [104–108]. Nevertheless, the addition of Ce increases the volatilization temperature of B$_2$O$_3$, leaving the fluidity of oxide scale largely unchanged. Therefore, the flow traces of glassy scale can still be observed on the MSBCe surface even at 1300 °C; in contrast, the MSB$_1$ surface has little flow traces (Figure 7c,f). There is no doubt that scale flow can heal pores and cracks on the alloy surface, thus Ce addition has a positive effect on the oxidation protection of MSB$_1$.

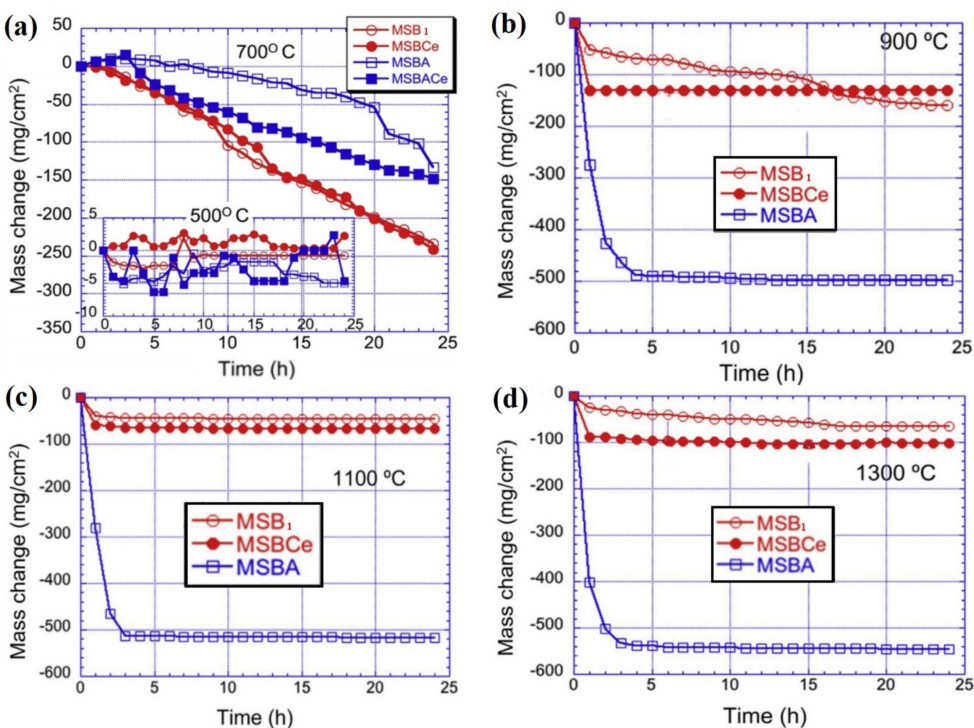

**Figure 6.** Mass loss curves of various alloys at different temperatures. (**a**) and (**b–d**) reproduced with permission from [102] and [103], respectively. Copyright 2010 Elsevier and 2016 Elsevier.

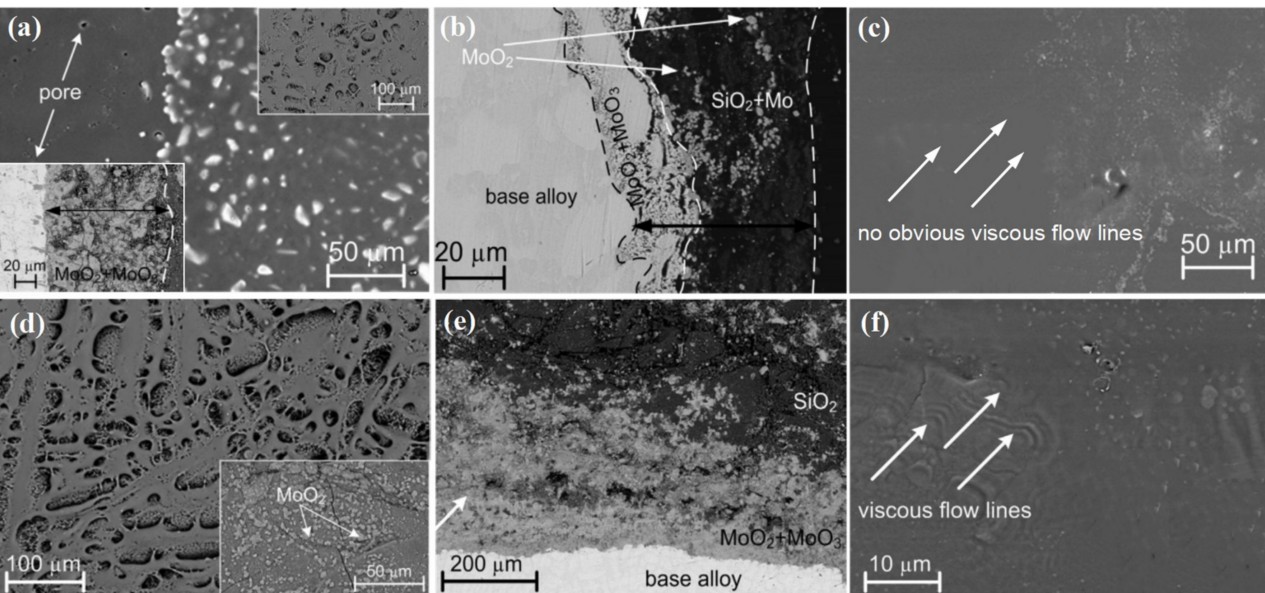

**Figure 7.** Surface and cross-sectional SEM (BSE) images of MSB$_1$ (**a–c**) and MSBCe (**d–f**) alloys oxidized at different temperatures for 24 h: 900 °C (**a,d**), 1100 °C (**b,e**) and 1300 °C (**c,f**). Reproduced with permission [103]. Copyright 2016 Elsevier.

In addition, Das et al. [109] also researched the oxidation reaction of Mo-Si-B-Al-Ce alloys at 1100 °C. It is well known that adding Al to Mo-Si-B systems may lead to the failure of alloy oxidation protection owing to the formation of mullite [110–114], which is also verified by the mass loss curves of Al-doped alloy in Figure 6b–d. Das [109] noted that adding Ce can further inhibit the malignant oxidation of Mo-Si-B-Al systems at 1100 °C, since the presence of Ce hindered the generation of mullite and promoted the formation of dense protective Al-oxide films on the alloy surface, thereby improving the antioxidant capacity.

#### 2.1.4. Effects of Y Element Addition

It has been sure that adding a Y element can significantly prolong the service life of Mo-based alloys due to the fact that Y will exhibit higher oxygen affinity than Mo [115,116]. Moreover, Y can also inhibit ion diffusion in grain boundaries and decrease the oxide-scale growth rate [117–119]. The presence of Y improves the adhesion between oxide film and substrate, thus improving the alloyed oxidation resistance [120,121]. Therefore, researchers try to add the Y element to Mo-Si-B alloys to obtain materials with better performance. After comparing the variation of Y-doped and Y-free Mo-9Si-8B (at.%) samples in the oxidation behavior at 650–1400 °C, Majumdar et al. [122–124] found that all samples presented a trend of transient mass increase followed by continuous rapid decrease at 650 °C (Figure 8a). Among all the samples, the 2 at.% Y-doped sample had the minimum mass loss at 750–1000 °C (Figure 8b–e), and the 0.2 at.% Y-doped sample presented the lowest mass loss at 1100 °C (Figure 8f). This phenomenon is attributed to the stable $Y_6MoO_{12}$ and $Y_5Mo_2O_{12}$ oxides produced in the initial period of oxidation at 750–1000 °C. These stable yttrium-molybdate oxides were formed rapidly, which inhibited the generation and vaporization of $MoO_3$, leading to a decrease in the mass loss of 2 at.% Y-doped sample. In addition, these yttrium-molybdate oxides often exist in the form of fine particles, which can accelerate the nucleation and growth of outer protective $SiO_2$ films.

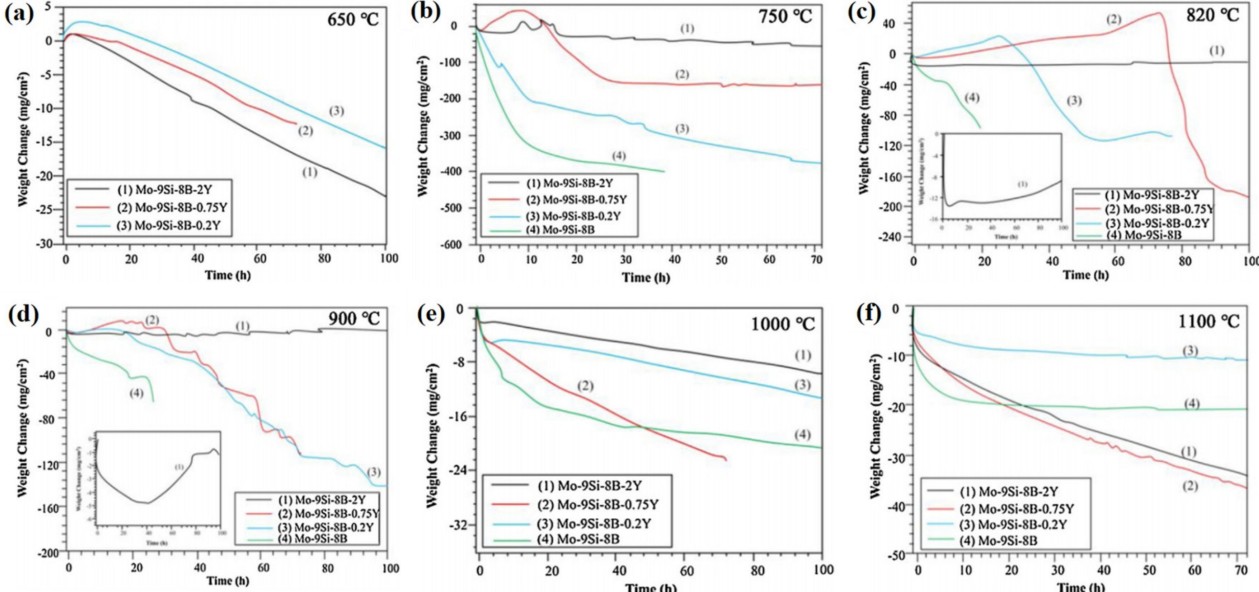

**Figure 8.** Weight change curves of Y-doped and undoped samples oxidized at (**a**) 650 °C; (**b**) 750 °C; (**c**) 820 °C; (**d**) 900 °C; (**e**) 1000 °C; (**f**) 1100 °C, respectively. Reproduced with permission [122]. Copyright 2014 Elsevier.

In order to further clarify the role of Y doping on the sample antioxidation behavior, the microstructure of oxidized samples are analyzed in detail [58,122]. Figure 9a,b give the cross-sectional micrographs of 2 at.% and 0.75 at.% Y-doped samples oxidized at 900 °C for 24 h, respectively. It can be seen that 0.75 at.% Y-doped sample has a thicker inner $MoO_2$ scale than that of 2 at.% Y-doped sample, which indicates that properly increasing the concentration of Y can inhibit the formation of $MoO_2$ to some extent. Furthermore, microcracks are also observed on the outer $SiO_2$ scale of 0.75 at.% Y-doped sample due to the quite high growth velocity (about 7.6 $\mu m \cdot h^{-1}$) of inner $MoO_2$ scale, which produces such a large tensile stress that outer $SiO_2$ scale breaks (Figure 9b,d). It is worth noting that the thickness of inner $MoO_2$ scale for 0.75 at.% Y-doped sample is significantly thinner after oxidation at 1100 °C for 72 h (Figure 9c). As $SiO_2$ will present viscous flow to cover holes and cracks on the alloy surface during the temperature exceeds 965 °C [125,126], which prevents further oxidation of the substrate. Figure 9e shows the thickness changes of $SiO_2$ and $MoO_2$ layers for 0.2 at.% Y-dpoed sample at 1100 and 1200 °C, which further

supports the above analysis results. When the oxidation temperature is higher than 1200 °C, a thin yttrium-silicate ($Y_2Si_2O_7$) scale is observed on the outer surface of Y-doped samples (Figure 10a–d), and the thickness of yttrium-silicate film gradually increases as the oxidation temperature raises (Figure 10c,e) [123]. It has been proven that the outer yttrium-silicate film is conducive to the alloy oxidation protection through preventing $SiO_2$ from forming volatile silicon hydroxide in humid conditions above 1200 °C [127–129]. Similar studies have been reported by Gorr et al. [130].

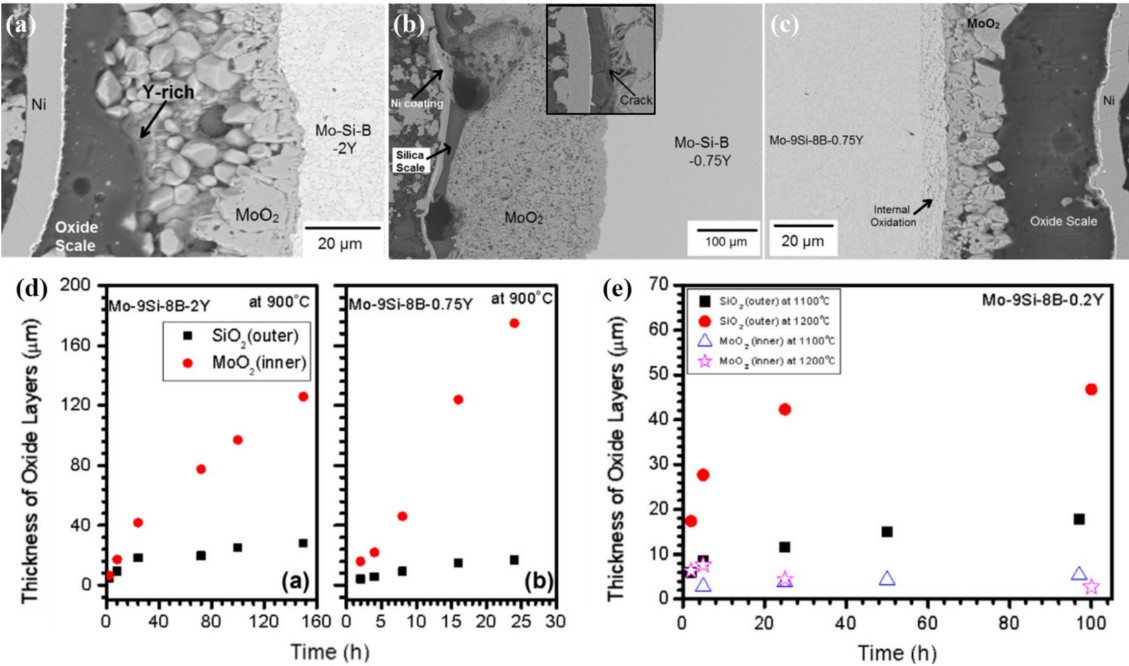

**Figure 9.** Cross-sectional BSE images of 2 at.% and 0.75 at.% Y-doped samples oxidized at differernt conditions: (**a**,**b**) 900 °C for 24 h, (**c**) 1100 °C for 72 h; (**d**,**e**) Changes of $MoO_2$ laye and $SiO_2$ layer thickness in Y-doped samples oxidized at different temperatures. Reproduced with permission [122]. Copyright 2014 Elsevier.

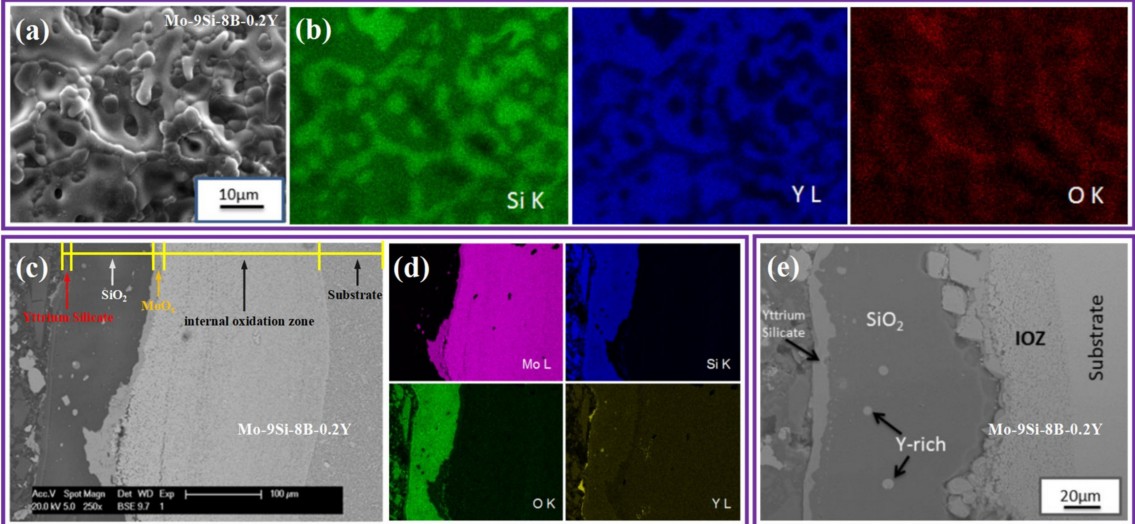

**Figure 10.** Surface SE (**a**) and cross-sectional BSE (**c**,**e**) images of 0.2 at.% Y-doped samples oxidized at different conditions: 1300 °C for 100 h (**a**,**c**) and 1400 °C for 2 h (**e**); (**b**,**d**) are the element mappings of (**a**,**c**), respectively. Reproduced with permission [123]. Copyright 2013 Springer Nature.

There is no doubt that alloying with Zr has a great impact on the antioxidant ability of Mo-Si-B materials. This is because the addition of Zr may produce polymorphic $ZrO_2$

or monomorphic $ZrSiO_4$, which mainly depends on the oxidation temperature. Among them, the $ZrSiO_4$ can act as an obstacle phase, which is beneficial to improve the alloy oxidation behavior; whereas the $ZrO_2$ will expand in volume at high temperatures (>1200 °C), which destroys the integrity of the $SiO_2$ scale so that it loses the protective effect [131,132]. Therefore, inhibiting the formation of $ZrO_2$ phase is essential to improve the alloy oxidation resistance.

Based on the fact that yttria suppresses the zirconia phase transition [133,134], Yang et al. [135] designed and fabricated Mo-12Si-10B-1Zr-0.3Y, Mo-12Si-10B-1Zr and Mo-12Si-10B samples (at.%, abbreviated as 1Zr-0.3Y, 1Zr-0Y and 0Zr-0Y, respectively). Figure 11a shows the mass variation of the three samples at 1250 °C, it can be seen that adding 1 at.% Zr to the 0Zr-0Y sample will lead to continuous and sharp mass loss. As the 0Zr-0Y sample has formed dense protective $SiO_2$ films during the oxidation, it avoids the sample sustained mass loss (Figure 12a), whereas the addition of Zr causes the $SiO_2$ scale to become loose and porous due to the formation of $ZrO_2$, and the porous structure provides channels for $O_2$ inward diffusion, thus accelerating the sample oxidation (Figure 12b). It is encouraging that further adding 0.3 at.% Y can effectively prevent the rapid mass loss of the 1Zr-0Y sample. As can be seen from Figure 12c, $ZrSiO_4$ rather than $ZrO_2$ appears on the sample surface after the addition of Y, thus eliminating the adverse effect of Zr doping. Meanwhile, the Y-Mo-rich oxide is also observed around the $ZrSiO_4$ phase. EDS analysis shows that the Y/Mo atomic ratio of this oxide is about 1/2, revealing that the oxide may be $Y_2Mo_4O_{15}$. Again, the XPS spectra also presents that the oxide has nearly the same Mo 3d and Y 3d bonding energies as $Y_2Mo_4O_{15}$ gauged through You et al. [136], which further verifies the above inference (Figure 11b,c). Furthermore, the 1Zr-0.3Y sample surface also forms a uniformly dense outer $Y_2Si_2O_7$ scale with the increase of oxidation time, which provides a better protection effect than 0Zr-0Y sample (Figure 12d). It has been observed from the cross-section enlarged Figure 12e,f that Y diffuses outward with the metastable $Y_2Mo_4O_{15}$ as the carrier and produces $Y_2Si_2O_7$ after a series of reactions at the top of $SiO_2$ scale, which will be accumulated and compressed to form the outer $Y_2Si_2O_7$ layer. Therefore, 1Zr-0.3Y sample presents the best antioxidant performance among the three samples.

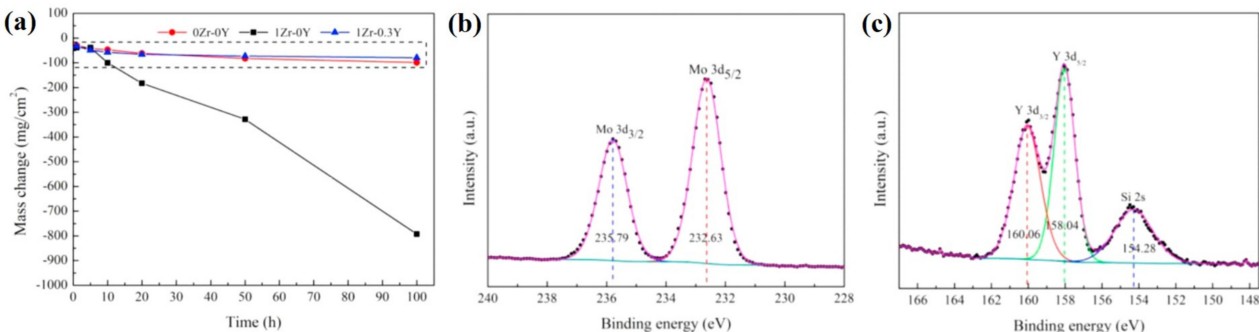

**Figure 11.** Mass change curves of three studied samples at 1250 °C (**a**); XPS analysis of Mo (**b**) and Y (**c**) characterized the 1Zr-0.3Y sample surface oxidized for 1 h. Reproduced with permission [135]. Copyright 2020 Elsevier.

### 2.2. Effects of Rare Earth Elements on Other Mo-Si Alloys

Previous studies have pointed out that adding Nb to the Mo-Si-based materials can play a satisfactory effect in improving mechanical properties due to damaging the stability of the $Mo_3Si$ phase [137–139]. However, the presence of Nb will lead to catastrophic oxidation of the material [140–142]. Inspired by the above study that adding Y can enhance the antioxidant properties of Zr-doped Mo-Si-B alloys, we further discussed the role of adding Y on the oxidation behavior of Nb-doped Mo-Si alloys.

Majumdar [143] used the non-consumable arc-melted method to prepare the undoped and 0.5Y-doped Mo-26Nb-19Si samples (at.%), which are simply referred to as Alloy1 and Alloy2, respectively. The microstructures of both samples are shown in Figure 13a,e. It can be found that they are both composed of dark and bright areas. According to XRD analysis

(Figure 14a) and EBSD mappings (Figure 13b–d,f–h), the dark and bright areas are (Mo, Nb)$_5$Si$_3$ and (Mo, Nb)$_{ss}$ phases, respectively. Moreover, Y$_2$O$_3$ particles are also observed on the Alloy2 grain boundaries. These particles can suppress the elongated grain growths, which results in the difference of microstructure morphology between the two samples. Meanwhile, Majumdar [143] also studied the oxidation process of Alloy2 at 1000 and 1300 °C. It is established that the sample exhibits continuous linear mass loss when oxidized at 1000 °C. When the oxidation temperature increases to 1300 °C, the sample is oxidized more vigorously and loses its antioxidant capacity within 2 h of oxidation, as shown in Figure 14b. Figure 15 shows the cross-section and surface micrographs of the oxidized sample. It can be discovered that the Alloy2 surface has formed a thick oxide layer after oxidation at 1000 °C for 24 h (Figure 15a). As can be seen from Figure 15d, the oxide layer is mainly composed of MoO$_2$, Nb$_2$O$_5$ and SiO$_2$, wherein Nb$_2$O$_5$ can act as a channel for O$_2$ internal diffusion due to the lack of protective action, which leads to rapid oxidation of the sample. In addition, the sample surface oxide film, which consists of Y$_2$O$_3$, Nb$_2$O$_5$ and SiO$_2$, appears numerous cracks and holes during oxidation at 1300 °C for 2 h (Figure 15b,c), resulting in the loss of protection from oxidation. Therefore, adding Y to Mo-Si-Nb alloys cannot overcome the oxidizing problem.

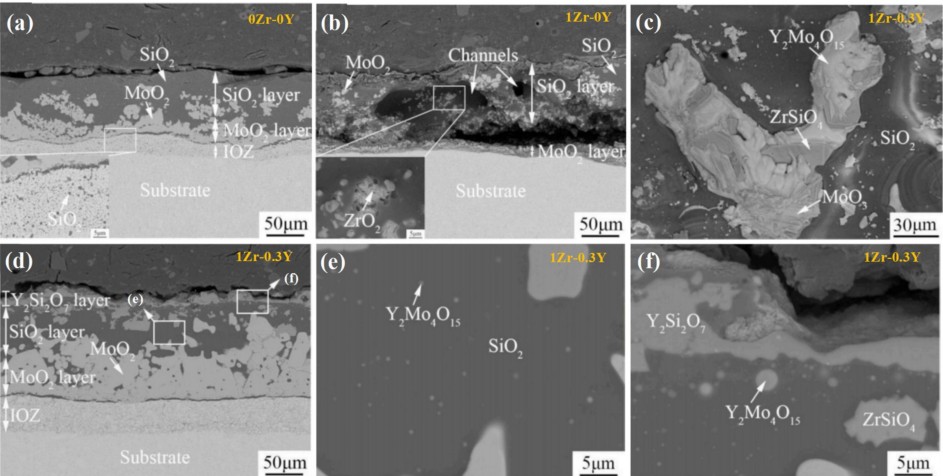

**Figure 12.** Cross-sectional and surface BSE images of three studied samples oxidized at 1250 °C for different time: (**a**) 20 h, (**b**) 5 h, (**c**) 1 h, (**d–f**) 50 h. Reproduced with permission [135]. Copyright 2020 Elsevier.

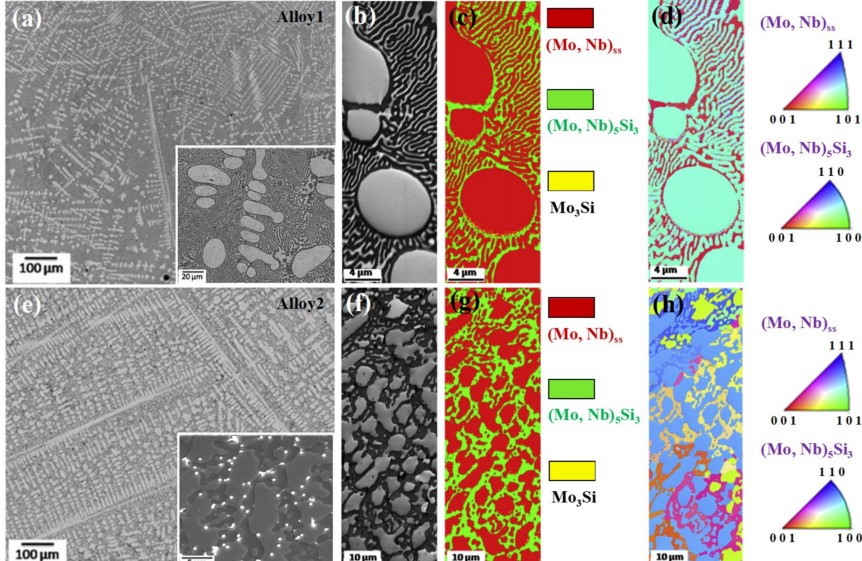

**Figure 13.** Microstructure morphologies of Alloy1 (**a**) and Alloy2 (**e**); EBSD mappings of Alloy1 and Alloy2: band contrast (**b,f**), phase (**c,g**), IPF (**d,h**). Reproduced with permission [143]. Copyright 2018 Elsevier.

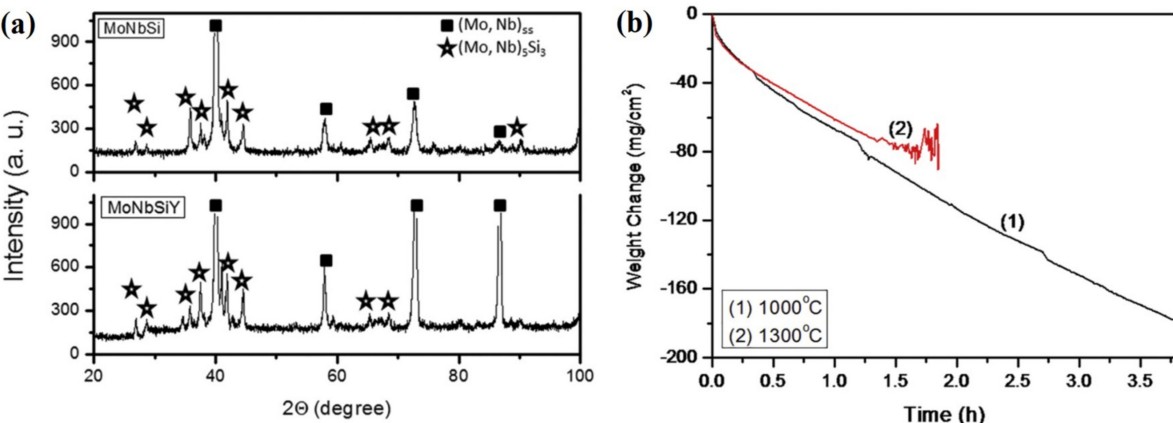

**Figure 14.** (**a**) XRD patterns of both samples before the oxidation; (**b**) weight change curves of Alloy2 oxidized at 1000 and 1300 °C, respectively. Reproduced with permission [143]. Copyright 2018 Elsevier.

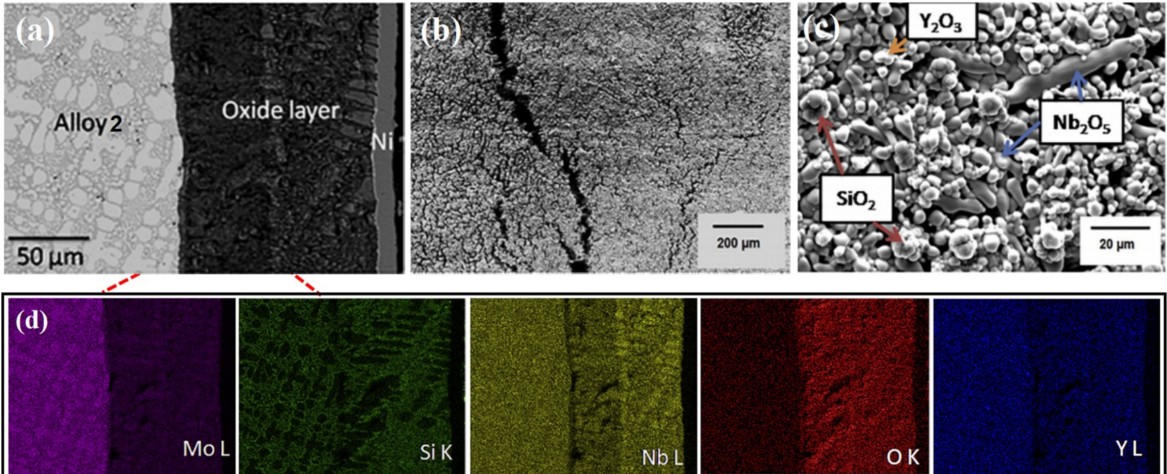

**Figure 15.** Cross-section BSE micrograph of Alloy2 (**a**) with the corresponding EDS mappings (**d**) oxidized at 1000 °C for 24 h; surface SEM/SE images of low (**b**) and high magnifications (**c**) for Alloy2 oxidized at 1300 °C for 2 h. Reproduced with permission [143]. Copyright 2018 Elsevier.

## 3. Strengthening Mechanism of Rare Earth Elements

According to the above research, it can be determined that the improvement of oxidation behavior of Mo-Si-based alloys by rare earth elements is mainly achieved through the following three ways. First, optimizing the alloy microstructure by refining grains or uniformly distributing phases, which is conducive to the rapid formation of protective oxide films on the alloy surface [80,96,103,144]. Second, producing stable rare earth oxides, these oxides are dispersed in scale and act as obstacle phases or diffusion barriers, which is conducive to suppressing the $MoO_3$ volatilization and $O_2$ inward diffusion [59,122,123]. Third, forming an additional rare earth oxide layer, thus further improving the antioxidant capacity [96,135].

However, it is disappointing that sometimes the addition of rare earth elements may even lead to the deterioration of alloy oxidation behavior in high temperature environments. For example, adding La to the Mo-Si-B system above 1100 °C has leaded to its accelerated oxidation attribute to the formation of large amounts of cracks and holes [60]. Therefore, the challenges ahead remain severe.

Figure 16 shows a schematic diagram of the oxidation process for rare earth element doped and undoped Mo-Si-based alloys at medium-high temperatures, which is helpful to further understand the strengthening mechanism of rare earth elements. It can be seen that the alloy with finer grain size can be prepared after adding rare earth elements like La,

which will affect the oxidation behavior to some extent. Overall, the oxidation process of the two kinds of alloys can be divided into two stages: initial and stable oxidation stages. During the initial oxidation stage, a discontinuous $SiO_2$ scale is formed on the surface of alloy without rare earth doping, which cannot effectively isolate oxygen. As a result, the alloy is oxidized violently and forms a Mo-oxide ($MoO_2$ and $MoO_3$) layer below the $SiO_2$ scale. Among them, $MoO_3$ is highly volatile, which leads to a severe mass loss of the alloy and leaves some holes and cavities on the surface [145]. Fortunately, $SiO_2$ gradually increases and flows to heal these holes and cavities as the oxidation time increases, thus facilitating the formation of continuous $SiO_2$ scale [146]. During the stable oxidation stage, the complete scale can provide sufficient protection for the substrate due to the effective restriction of $O_2$ internal diffusion, resulting in the reduction of oxygen pressure inside the alloy. Obviously, low oxygen partial pressure inhibits the continuous generation of $MoO_2$, and the original $MoO_2$ will continue to oxidize to produce $MoO_3$ and slowly volatilize so that the Mo-oxide interlayer becomes thinner [147]. Meanwhile, the substrate below the $MoO_2$ layer has been oxidized selectively, leading to the emergence of an internal oxidation zone [122], as shown in Figure 16a. In contrast, the alloy doped with rare earth can generate rare earth oxides such as $La_2O_3$, $Y_6MoO_{12}$, $Y_5Mo_2O_{12}$, etc., in the initial oxidation stage. These stable oxides, on the one hand, promote the formation of continuous $SiO_2$ scale. On the other hand, they fill holes in the scale to eliminate the shortcut of $O_2$ inward diffusion and $MoO_3$ volatilization, so that the alloy can enter the stable oxidation stage faster. In addition, a double-layer protective oxide film (i.e., $Y_2Si_2O_7$-$SiO_2$ or $SiO_2$-$HfSiO_4$ duplex scales) is formed on the alloy surface during the stable oxidation stage, providing more effective protection against oxidation, as shown in Figure 16b.

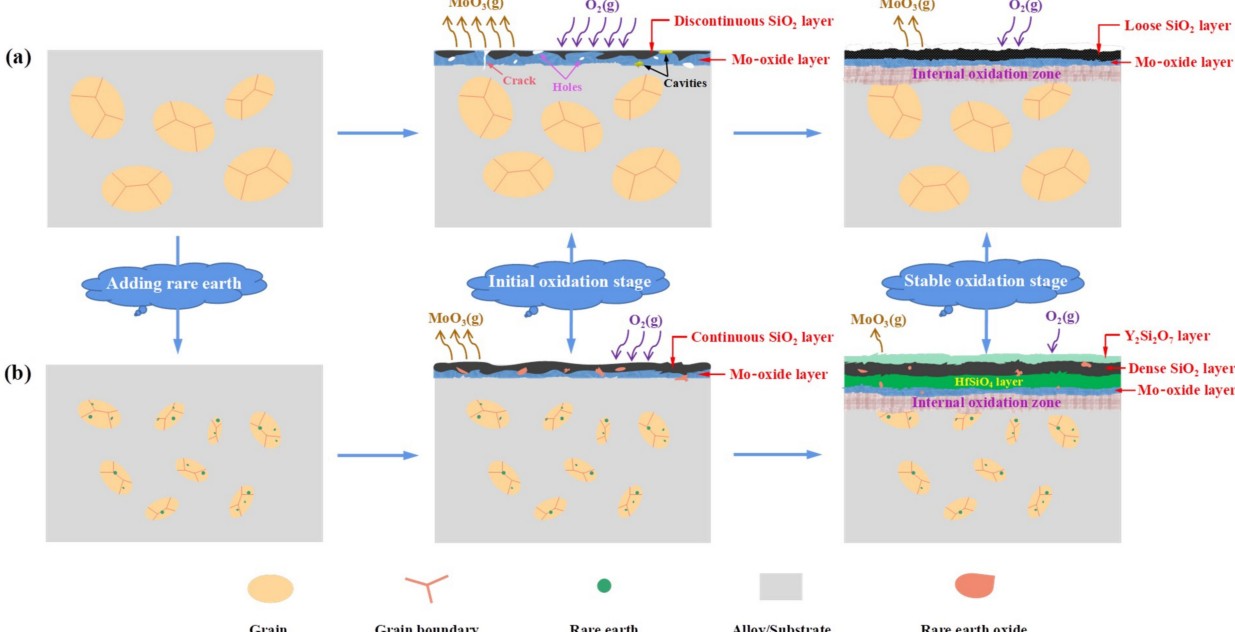

**Figure 16.** Schematic diagram of the oxidation process for Mo-Si-based alloy at medium-high temperature: (**a**) without rare earth elements, (**b**) doped with rare earth elements.

## 4. Conclusions and Outlook

This paper reviewed the role of rare earth elements on the oxidation behavior of Mo-Si-based alloys, and summarized the strengthening mechanism of various rare earth elements. Adding La to Mo-Si-B alloys can make grains become finer and produce stable La-containing oxides with a "pinning" effect, which significantly enhances the oxidation resistance below 1000 °C. Mo-Si-B alloys doped with Hf can generate $HfSiO_4$ particles that promote the healing of holes and cracks in oxide scales; besides, it is likely that a $HfSiO_4$ inner layer will be formed to inhibit the $MoO_3$ volatilization and $O_2$ inward diffusion.

Alloying with Ce can shorten the transient oxidation period of Mo-Si-B alloys, meanwhile it also raises the volatilization temperature of $B_2O_3$ in oxide films, which is conducive to maintaining the viscosity and integrity of borosilicate scale. Moreover, adding Ce to the Mo-Si-B-Al system can also hinder the formation of mullite and promote the emergence of protective Al-oxide scale, which can provide more effective protection against oxidation. Adding Y to Mo-Si-B alloys will produce stable $Y_6MoO_{12}$ and $Y_5Mo_2O_{12}$ oxides or create a $Y_2Si_2O_7$ outer layer that can act as diffusion barriers. Furthermore, adding Y to the Mo-Si-B-Zr system also suppresses $ZrO_2$ formation, thus eliminating the adverse effect of Zr doping on oxidation behavior.

However, it is noteworthy that adding rare earth elements do not always improve the antioxidation ability of Mo-Si-based alloys. For example, adding Y to the Mo-Si-Nb system cannot prevent its catastrophic oxidation; adding La also leads to accelerated oxidation of Mo-Si-B alloys above 1000 °C. Therefore, further research for other oxidation protection methods is necessary. Some research schemes worth exploring in the future are listed below. Before using, preoxidation treatment at an appropriate temperature can obtain protective silica scales on the alloy surface, thus effectively inhibiting the inward diffusion of $O_2$ and obviously extending the service life of alloy. Processing of preceramic polymers is a very promising method owing to its simple operation and low cost. Establishing a relevant numerical simulation or mathematical model to quantitatively study the relationship between oxidation behavior and microstructure. Combined with the emerging coating technology to design a suitable silicide-based coating such as $MoSi_2$ ceramic coating for the Mo-Si-based system.

**Author Contributions:** The manuscript was written through the contributions of all authors. Y.Z.: conceptualization, investigation, and supervision. Y.Z. and L.Y.: writing—original draft and image processing. L.Y., T.F., K.C. and F.S.: validation, resources, investigation, writing—review and editing. Y.Z., L.Y. and J.W.: visualization, Writing—review and editing. All authors have read and agreed to the published version of the manuscript.

**Funding:** This research was funded by the National Natural Science Foundation of China, Grant No. 51604049.

**Institutional Review Board Statement:** Not applicable.

**Informed Consent Statement:** Not applicable.

**Data Availability Statement:** Not applicable.

**Acknowledgments:** This work was supported by the National Natural Science Foundation of China (No. 51604049).

**Conflicts of Interest:** The authors declare no conflict of interest.

**Notes:** The authors declare no competing financial interest.

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
