# Peer review of "Rare Earth Elements Enhanced the Oxidation Resistance of Mo-Si-Based Alloys for High Temperature Application: A Review"

_coatings, doi:10.3390/coatings11091144_

Round 1
Reviewer 1 Report
The manuscript is devoted to the influence of rare earth elements addition ( La, Hf, Ce and Y ) on the microstructure and oxidation resistance of Mo-Si-based alloys. The paper presents a review of selected works devoted to the resistance to oxidation of Mo-Si, Mo-Si-B and Mo-Si-X alloys at elevated temperatures, and changes in this resistance as a result of the addition of rare earth elements. The main content of the work is a description of the results of the work published in the above-mentioned subject supported by the authors' comments, which is typical for reviews.
The work requires introducing some changes and additions before publication. Comments and comments concern:
1. abstract gives the purpose of the work and motivations for the research undertaken. Lack of the main results of the analysis or conclusions.
2. introduction line 58. wrong section number, change 1 into 2, Before discussing the influence of rare earth elements, it is worth presenting the role of alloying elements (Si, B) on the properties of the Mo-Si alloy. It is recommened to place phase diagrams for Mo-Si, Mo-Si-B and Mo-Si-X (ternary or pseudobinary) in the appropriate sections of the review.line 66-68 redundant sentence, the aim of the work was given in abstract and in the introduction, too.
3. Figure 2, too many small figures, markings and letters are so small(fig.2d), it’s hard to recognize.What is presented on color figures placed next to the SEM image? EDS linescans for elements are given at the SEM image ( without scales and units)?Please unify letters size and font as well as scale bars and numbers and their positions on figures 2d-g.
4. Table 2 and Figures 3-5 It is recommended to place tables and figures right after the sentence/ text paragraph they are mentioned, not in the end of the section. This remark also applies to the remaining paragraphs.
5. Figure 3, please unify scale bars position and letters ( size and font). Figures 3g, h, i – are poor quality images.
6. Figure 5b,Please identify the position for EDS analysis, markers of identified elements are too small thus almost invisibleon presented EDS spectra.
7. Figure 6, Please unify the name of the timeline. Figures 6b and d, what the black arrows in the charts mean?
8. What important is presented in figure 7c, add a comment or markings, or delete the image.
9. Figures 13 c,d, g and h- The letters in the descriptions are too smallfigure 15 d ( EDS maps for O and Y, no contrast, they show nothing as to why post them?
10. I believe that paragraph 3, describing the reinforcement mechanisms, should come before paragraph 2.
11. Conclusions - it is rather a summary of the analysis of selected works and they should be defined in this way
Author Response
Dear Editors and Reviewers:
Thank you for your letter and for the reviewers’ comments concerning our manuscript entitled “Rare earth elements enhanced the oxidation resistance of Mo-Si-based alloys for high temperature application: A review” (ID: coatings-1378439). Those comments are all valuable and very helpful for revising and improving our paper, as well as the important guiding significance to our researches. We have studied comments carefully and have made correction which we hope meet with approval. Revised portion are marked in red in the paper. The main corrections in the paper and the responds to the reviewers’ comments are as following:
Responds to the Editor and reviewer’s comments:
Reviewer #1:
- Abstract gives the purpose of the work and motivations for the research undertaken. Lack of the main results of the analysis or conclusions.
Thank you very much for your comments. We have added some the main results of the analysis or conclusions in the abstract section, and sincerely hope that our modifications can be approved by you.
- Introduction line 58. wrong section number, change 1 into 2, Before discussing the influence of rare earth elements, it is worth presenting the role of alloying elements (Si, B) on the properties of the Mo-Si alloy. It is recommended to place phase diagrams for Mo-Si, Mo-Si-B and Mo-Si-X (ternary or pseudobinary) in the appropriate sections of the review. Line 66-68 redundant sentence, the aim of the work was given in abstract and in the introduction, too.
Thank you very much for your comments. We have made corresponding modifications in the review. Such as the wrong title serial number is modified, the effects of Si and B elements on the properties of the alloy are added in the introduction, the phase diagram of Mo-Si-B ternary alloy is added in Fig. 1 and redundant sentences in the 2 section were deleted. Sincerely hope that our modification can be recognized by you.
- Figure 2, too many small figures, markings and letters are so small (Fig.2d), it’s hard to recognize. What is presented on color figures placed next to the SEM image? EDS linescans for elements are given at the SEM image ( without scales and units)? Please unify letters size and font as well as scale bars and numbers and their positions on figures 2d-g.
Thank you very much for your comments. We have modified Fig. 2 in detail. For example, redundant small numbers and pictures (like the EDS line-scans for elements are given at the SEM image, and color figures placed next to the SEM image) are deleted, and the size, font and position of letters are unified. Sincerely hope that our modification can be recognized by you.
- Table 2 and Figures 3-5: It is recommended to place tables and figures right after the sentence/text paragraph they are mentioned, not in the end of the section. This remark also applies to the remaining paragraphs.
Thank you very much for your comments. We have made appropriate adjustments to the position of the tables and figures in the review. Sincerely hope that our modification can be recognized by you.
- Figure 3, please unify scale bars position and letters (size and font). Figures 3g, h, i – are poor quality images.
Thank you very much for your comments. We have modified Fig. 3 in detail. For example, the position of the scale bars (located in the lower right corner of each small picture) and letters (size and font) are unified, and the images (Fig. 3g-i) with poor quality are deleted. Sincerely hope that our modification can be recognized by you.
- Figure 5b, Please identify the position for EDS analysis, markers of identified elements are too small thus almost invisible on presented EDS spectra.
Thank you very much for your comments. We have made corresponding modifications in Fig. 5 to solve this problem, such as enlarging the markers, etc. Sincerely hope that our modification can be recognized by you.
- Figure 6, Please unify the name of the timeline. Figures 6b and d, what the black arrows in the charts mean?
Thank you very much for your comments. We have unified the name of the timeline in Fig. 6. The specific significance represented by the black arrows in Figs. 6b and d is described in detail in reference [103], but it is not essential to study its significance in depth in this review. So we deleted the black arrows in the Figs. 6b and d. Sincerely hope that our modification can be recognized by you.
- What important is presented in figure 7c, add a comment or markings, or delete the image.
Thank you very much for your comments. We have added some markings in Fig. 7c for comparison with Fig. 7f to reveal the effect of Ce addition on the Mo-Si-B alloys. Sincerely hope that our modification can be recognized by you.
- Figures 13 c, d, g and h- The letters in the descriptions are too small. Figure 15 d ( EDS maps for O and Y, no contrast, they show nothing as to why post them?
Thank you very much for your comments. We have adjusted the size of letters in the Fig. 13 and the contrast of O (red) and Y (blue) in the Fig. 15 d. Besides, the Fig. 15 d is placed to better illustrate the composition of the oxide layer formed on the surface of alloy2. Sincerely hope that our modification can be recognized by you.
- I believe that paragraph 3, describing the reinforcement mechanisms, should come before paragraph 2.
Thank you very much for your comments. We have adjusted the position of the corresponding paragraph, and sincerely hope that our modification can be recognized by you.
- Conclusions - it is rather a summary of the analysis of selected works and they should be defined in this way
Thank you very much for your comments. We have made a comprehensive modifications to the conclusion. Sincerely hope that our modifications can be approved by you.
Finally, thank you again for your valuable suggestions, which make our article more perfect. We sincerely hope that our modifications can be recognized by you.
Reviewer 2 Report
The authors provide a review of the rare earth elements enhanced the oxidation resistance of Mo-Si-based alloys for high temperature application. The paper can be of interest for Coatings. However, VERY MAJOR revisions are requested:
- There part dealing with the mechanical behavior is too small. The authors must provide a more detailed discussion based on nanoindentation and advanced test including micropillar compression and splitting. The authors should include other paper focusing on the mechanical properties of ceramics at the submicrometre scale, doi.org/10.1111/jace.15093, doi.org/10.1016/j.matdes.2019.107762 doi.org/10.1063/1.3595423, doi.org/10.3139/146.110149 and doi.org/10.1016/j.matdes.2018.04.009. This is a key point to be addressed a ceramic resistant to oxidation without good mechanical properties will have poor applications impact.
- Amorphous ceramics and composites in thin films are known for their good mechanical properties especially at the sub micrometer scale in thin film form or ribbon as reported in doi.org/10.1016/j.actamat.2021.116955, doi.org/10.1063/1.125654, doi.org/10.1016/j.matdes.2018.107565 for the case of ZrCuO alloys. The authors must comment on this paper developing a section devoted to ceramic thin films and their mechanical properties. This make the review more appropriate for coatings
- The authors must also discuss in a better why the amorphization phenomena and of Mo-Si ceramics This should consider both the effect of the cooling rate and composition
- The quality of Figure 4 must be improved.
Author Response
Dear Editors and Reviewers:
Thank you for your letter and for the reviewers’ comments concerning our manuscript entitled “Rare earth elements enhanced the oxidation resistance of Mo-Si-based alloys for high temperature application: A review” (ID: coatings-1378439). Those comments are all valuable and very helpful for revising and improving our paper, as well as the important guiding significance to our researches. We have studied comments carefully and have made correction which we hope meet with approval. Revised portion are marked in blue in the paper. The main corrections in the paper and the responds to the reviewers’ comments are as following:
Responds to the Editor and reviewer’s comments:
Reviewer #2:
- There part dealing with the mechanical behavior is too small. The authors must provide a more detailed discussion based on nanoindentation and advanced test including micropillar compression and splitting. The authors should include other paper focusing on the mechanical properties of ceramics at the submicrometre scale, doi.org/10.1111/jace.15093, doi.org/10.1016/j.matdes.2019.107762, doi.org/10.1063/1.3595423, doi.org/10.3139/146.110149 and doi.org/10.1016/j.matdes.2018.04.009. This is a key point to be addressed a ceramic resistant to oxidation without good mechanical properties will have poor applications impact.
Thank you very much for your comments. There have been many and comprehensive studies on the mechanical behavior of Mo-Si based alloys, so we mainly studied the oxidation behavior of Mo-Si based alloys in this review. For their mechanical properties, we mainly made a brief introduction in the introduction. In addition, we also added some comments on the mechanical properties of other ceramic materials and cited these recommended references (placed in references [3-7]). Sincerely hope that our modifications can be approved by you.
- Amorphous ceramics and composites in thin films are known for their good mechanical properties especially at the sub micrometer scale in thin film form or ribbon as reported in doi.org/10.1016/j.actamat.2021.116955, doi.org/10.1063/1.125654, doi.org/10.1016/j.matdes.2018.107565 for the case of ZrCuO alloys. The authors must comment on this paper developing a section devoted to ceramic thin films and their mechanical properties. This make the review more appropriate for coatings.
Thank you very much for your comments. We have added some comments on this paper developing a section devoted to ceramic thin films and their mechanical properties in the introduction and cited these recommended references (placed in references [23, 40, 41]). Sincerely hope that our modifications can be approved by you.
- The authors must also discuss in a better why the amorphization phenomena and of Mo-Si ceramics. This should consider both the effect of the cooling rate and composition.
Thank you very much for your comments. We have made corresponding modifications, such as adding some comments about the amorphization phenomena of Mo-Si ceramics in the section 2.1.2. Sincerely hope that our modifications can be approved by you.
- The quality of Figure 4 must be improved.
Thank you very much for your comments. We have improved the quality of Fig. 4, and sincerely hope that our modifications can be approved by you.
Finally, thank you again for your valuable suggestions, which make our article more perfect. We sincerely hope that our modifications can be recognized by you.
Reviewer 3 Report
Hello,
The manuscript "Rare earth elements enhanced the oxidation resistance of Mo-Si-based alloys for 2 high temperature application: A review” has been reviewed. It is an interesting topic, but this paper lacks focus, and does not provides a detailed discussion on the field by comparing the results of different alloys and samples. It looks mainly like a collection of data and results from literature and different papers, rather than raising the contradictory points, and aligning the discussion toward a promising direction. Therefore, a major revision before accepting the paper is recommended. Some more details/examples are provided as follow:
- As an example, the reason behind the following items has not been well elaborated (certainly, there are much more in the manuscript need further elaboration):
- According to 8, the alloy containing 2% Y presented the maximum weight loss at 650C, but it exhibited the minimum weight loss between 750-100C. And at 1100C, it again displayed the maximum weight loss.
There is no solid explanation what was the reason for this evolution
- “By the way, the weight gain of X341 sample is smaller than that of X342 sample owing to the fact that X342 sample presents special arrangement and finer grain size.”
- The Conclusion section is too large, and it is necessary to be shorten and written in a more concise and clear manner;
- The authors did not pay attention even for some basic requirements and formatting. The numeration of the Titles and Subtitles are not in a chronological order. See examples below:
"1. Introduction", "1. Effects of rare earth ...", "2.1. Effects of rare earth elements"
- The words used in title "1" have been repeated in "2.1" (they are exactly the same);
"1. Effects of rare earth elements on oxidation behavior of Mo-Si-based alloys"
"2.1. Effects of rare earth elements on oxidation behavior of Mo-Si-B alloys"
- In Fig.6, it has not been explained the term "MSBA" stands for what? "MSBA" has not been clarified neither in the Fig caption, nor in the text.
- The quality of some photos are very low (e.g., Fig 2), and it is necessary to edit them to be legible;
- It is necessary the authors to find a native English speaker to proofread the manuscript. There are a lot of grammatical errors in the text, which must be modified; for instance:
- In conlucion it has been written that: “Therefore, a further search for other oxidation”, it is better to be modified to “Therefore, a further research for other oxidation”;
- after semicolon, the word must be started with a "minuscule" word and not "capital"; example: “cracks in oxide scales; Besides, it is likely that”;
- Some more comments are available in the attached PDF file.
Please see attached file.
Cheers,

Author Response
Dear Editors and Reviewers:
Thank you for your letter and for the reviewers’ comments concerning our manuscript entitled “Rare earth elements enhanced the oxidation resistance of Mo-Si-based alloys for high temperature application: A review” (ID: coatings-1378439). Those comments are all valuable and very helpful for revising and improving our paper, as well as the important guiding significance to our researches. We have studied comments carefully and have made correction which we hope meet with approval. Revised portion are marked in green in the paper. The main corrections in the paper and the responds to the reviewers’ comments are as following:
Responds to the Editor and reviewer’s comments:
Reviewer #3:
- As an example, the reason behind the following items has not been well elaborated (certainly, there are much more in the manuscript need further elaboration): 1) According to 8, the alloy containing 2% Y presented the maximum weight loss at 650℃, but it exhibited the minimum weight loss between 750-1000℃. And at 1100℃, it again displayed the maximum weight loss. There is no solid explanation what was the reason for this evolution. 2) By the way, the weight gain of X341 sample is smaller than that of X342 sample owing to the fact that X342 sample presents special arrangement and finer grain size.
Thank you very much for your comments. We have added some comments in the review to elaborate the reasons behind some phenomena. For example, for question 1), we added comments (this phenomenon is attributed to the stable Y6MoO12 and Y5Mo2O12 oxides produced in the initial period of oxidation at 750-1000 ℃. These stable yttrium-molybdate oxides were formed rapidly, which inhibited the generation and vaporization of MoO3, leading to a decrease in the mass loss of 2at.% Y-doped sample. In addition, these yttrium-molybdate oxides often exist in the form of fine particles, which can accelerate the nucleation and growth of outer protective SiO2 films.) to explain the phenomenon of mass change of 2at.% Y sample, which is shown in section 2.1.4. For question 2), we also added some explanations (this is because HfB2 and MoB grains in the X342 sample present finer size and more uniform distribution than the X341 sample, which makes the more intense oxidation of X342 sample), which is shown in section 2.1.2. Sincerely hope that our modifications can be approved by you.
- The conclusion section is too large, and it is necessary to be shorten and written in a more concise and clear manner.
Thank you very much for your comments. We have made a comprehensive modifications to the conclusion. For example, simplify or shorten the meaning of cumbersome sentences and delete redundant sentences. Sincerely hope that our modifications can be approved by you.
- The authors did not pay attention even for some basic requirements and formatting. The numeration of the Titles and Subtitles are not in a chronological order. See examples below: "1. Introduction", "1. Effects of rare earth ...", "2.1. Effects of rare earth elements"
Thank you very much for your comments. We have modified the format in the Titles and Subtitles. For example, the serial numbers of Titles and Subtitles have been modified again in a chronological order. Sincerely hope that our modifications can be approved by you.
- The words used in title "1" have been repeated in "2.1" (they are exactly the same); "1. Effects of rare earth elements on oxidation behavior of Mo-Si-based alloys". "2.1. Effects of rare earth elements on oxidation behavior of Mo-Si-B alloys".
Thank you very much for your comments. We have modified the words in these two titles. For example, the two titles are changed to "2. Effects of rare earth elements on oxidation behavior" and "2.1. Effects of rare earth elements on Mo-Si-B alloys" respectively. Sincerely hope that our modifications can be approved by you.
- In Fig.6, it has not been explained the term "MSBA" stands for what? "MSBA" has not been clarified neither in the Fig caption, nor in the text.
Thank you very much for your comments. The term "MSBA" stands for "Mo-11.2Si-8.1B-7.3Al" alloy, and we have added this explanation in section 2.1.3. Sincerely hope that our modifications can be approved by you.
- The quality of some photos are very low (e.g., Fig 2), and it is necessary to edit them to be legible.
Thank you very much for your comments. We have adjusted the quality of some photos, such as Figs. 1, 2, 3, 4, etc. Sincerely hope that our modifications can be approved by you.
- It is necessary the authors to find a native English speaker to proofread the manuscript. There are a lot of grammatical errors in the text, which must be modified; for instance: In conlucion it has been written that: “Therefore, a further search for other oxidation”, it is better to be modified to “Therefore, a further research for other oxidation”.
Thank you very much for your comments. We have corrected some grammatical errors in the manuscript. Meantime, We have modified “Therefore, a further search for other oxidation” to “Therefore, a further research for other oxidation” in the conclusion. Sincerely hope that our modifications can be approved by you.
- After semicolon, the word must be started with a "minuscule" word and not "capital"; example: “cracks in oxide scales; Besides, it is likely that”.
Thank you very much for your comments. We have modified the word after the semicolon to start with a "minuscule" word in the review, and sincerely hope that our modifications can be approved by you.
- Some more comments are available in the attached PDF file.
Thank you very much for your comments. For the problems mentioned in the attached PDF file, we have modified them and marked them in the corresponding position in the review. Sincerely hope that our modifications can be approved by you.
Finally, thank you again for your valuable suggestions, which make our article more perfect. We sincerely hope that our modifications can be recognized by you.
Round 2
Reviewer 2 Report
-
Reviewer 3 Report
Thanks for applying the comments.